# Synthesis and Evaluation of Antimicrobial and Cytotoxic Activity of Oxathiine-Fused Quinone-Thioglucoside Conjugates of Substituted 1,4-Naphthoquinones

**DOI:** 10.3390/molecules25163577

**Published:** 2020-08-06

**Authors:** Yuri E. Sabutski, Ekaterina S. Menchinskaya, Ludmila S. Shevchenko, Ekaterina A. Chingizova, Artur R. Chingizov, Roman S. Popov, Vladimir A. Denisenko, Valery V. Mikhailov, Dmitry L. Aminin, Sergey G. Polonik

**Affiliations:** 1G.B. Elyakov Pacific Institute of Bioorganic Chemistry, Far-Eastern Branch of the Russian Academy of Sciences. Prospekt 100-let Vladivostoku 159, 690022 Vladivostok, Russia; ekaterinamenchinskaya@gmail.com (E.S.M.); lshev@piboc.dvo.ru (L.S.S.); martyyas@mail.ru (E.A.C.); chingizov84@gmail.com (A.R.C.); prs_90@mail.ru (R.S.P.); vladenis@piboc.dvo.ru (V.A.D.); vvmikhailov@inbox.ru (V.V.M.); daminin@piboc.dvo.ru (D.L.A.); 2Department of Biomedical Science and Environmental Biology, Kaohsiung Medical University, Kaohsiung 80708, Taiwan

**Keywords:** 1,4-naphthoquinones, quinoid compounds, thioglycosides, quinone-sugar conjugates, cytotoxic activity, antibiotic activity

## Abstract

A series of new tetracyclic oxathiine-fused quinone-thioglycoside conjugates based on biologically active 1,4-naphthoquinones and 1-mercapto derivatives of per-*O*-acetyl d-glucose, d-galactose, d-xylose, and l-arabinose have been synthesized, characterized, and evaluated for their cytotoxic and antimicrobial activities. Six tetracyclic conjugates bearing a hydroxyl group in naphthoquinone core showed high cytotoxic activity with EC_50_ values in the range of 0.3 to 0.9 μM for various types of cancer and normal cells and no hemolytic activity up to 25 μM. The antimicrobial activity of conjugates was screened against Gram-positive bacteria (*Staphylococcus aureus*, *Bacillus cereus*), Gram-negative bacteria (*Pseudomonas aeruginosa* and *Escherichia coli*), and fungus *Candida albicans* by the agar diffusion method. The most effective juglone conjugates with d-xylose or l-arabinose moiety and hydroxyl group at C-7 position of naphthoquinone core at concentration 10 µg/well showed antimicrobial activity comparable with antibiotics vancomicin and gentamicin against Gram-positive bacteria strains. In liquid media, juglone-arabinosidic tetracycles showed highest activity with MIC 6.25 µM. Thus, a positive effect of heterocyclization with mercaptosugars on cytotoxic and antimicrobial activity for group of 1,4-naphthoquinones was shown.

## 1. Introduction

Naphthoquinones represent one of the largest families of natural products and are widespread in nature. They were isolated from plants, marine invertebrates, fungi, and bacteria [1]. Naphthoquinones revealed a diverse spectrum of activities: anticancer [2,3,4], antibacterial [5,6], anti-infective [7], antimalarial [8], and cardioprotective action [9]. The quinone ring contains a system of double bonds conjugated with carbonyl groups: it is easily susceptible to reduction, oxidation, and addition of *O*-, *N*-, and *S*-nucleophiles [10,11]. The high reactivity of naphthoquinones and the well-developed methods of its chemical modification make this group of compounds attractive for the profound development of new types of substances with high biological activity [12].

Studies are continuing on the antimicrobial activity of 1,4-naphthoquinones in relation to various microbial pathogens that are dangerous as sources of fatal diseases, epidemics, and nosocomial infections. In some cases, not only was the direct effect of new compounds on microbial cells investigated, but also their effect on the viability of biofilms formed by reproducing microorganisms. Thus, a series of new 2-hydroxy-3-phenylsulfanylmethyl-1,4-naphthoquinones were synthesized and evaluated against Gram-negative and Gram-positive bacterial strains and their biofilms to probe for potential lead structures. The structure modification applied in the series resulted in 12 new naphthoquinones with pronounced antimicrobial activity against *Escherichia coli* and *Pseudomonas aeruginosa*. Four molecules showed anti-biofilm activity and inhibited biofilm formation more than 60% with a better profile than standard antibacterial drug, ciprofloxacin [13,14].

Naphthoquinones often possess poor solubility which has hampered their practical use. The conjugation of naphthoquinones with non-toxic carbohydrates is one of the most successful ways for improving their solubility [15,16,17,18,19]. Moreover, the conjugation of naphthoquinones with carbohydrates led to novel structures with new types of biological activity [20,21]. Such naphthoquinone–carbohydrate conjugates include classical *O*- and *S*-glycosides (carbohydrates linked directly to naphthoquinone via glycosidic bond), non-glycosidic conjugates (connection with the carbohydrate component via not glycosidic ether linkage), and *N*-glycosyl triazoles (a triazolic ring connecting the carbohydrate moiety to naphthoquinone). In the course of our drug research project we developed an effective method for preparation of naphthoquinone acetylthioglucosides by the condensation of available substituted chloromethoxynaphthoquinone **1** with per-*O*-acetyl-1-thioderivatives of d-glucose **2a**, d-galactose **2b**, d-xylose **2c**, and l-arabinose **2d** and obtained related naphthoquinone acetylglucosides **3a**–**d** [22]. These acetylglycoside derivatives, **3a**–**d**, were readily deacetylated with MeONa/MeOH and immediately converted to the quinone–sugar tetracyclic conjugates **4a**–**d** in good yields (Scheme 1). The tetracyclic quinone–carbohydrate conjugates **4a**–**d** had a linear planar structure and retained the stereochemistry of the starting sugars.

The obtained sugar–quinone tetracycles were converted to acetyl derivatives by treatment with Ac_2_O/Py. Both synthesized tetracyclic quinone conjugates and their acetylated tetracyclic derivatives were active in vitro against human promyelocytic leukemia HL-60 in 1.0–5.0 μM concentrations, while starting acyclic acetylglycosides were approximately 10–100 times less active [23].

## 2. Results and Discussion

The synthesis of tetracyclic oxathiine-fused glycoside naphthoquinone conjugates can provide bioactive compounds, and the variation of naphthoquinone and carbohydrate moieties allows a structure–activity relationship (SAR) study. Therefore, this work aimed to conjugate per-*O*-acetylated 1-thiosugars **2a**–**d** with various substituted 1,4-naphthoquinones. The key intermediates, per-*O*-acetylated 1-thiosugars **2a**–**d**, were prepared from the respective peracetylated glycosyl halides (d-glucose, d-galactose, d-xylose, and l-arabinose) using reducing cleavage of its thiouronium salt with sodium metabisulfite according to literature procedures [24,25,26]. Sugar thioderivatives comprise two pairs of structurally related carbohydrates: hexopyranoses of d-glucose and d-galactose, as well as pentopyranoses of d-xylose and l-arabinose, which differ in the configuration of the C_4_-OH group.

### 2.1. Preparation of Starting Chloro(bromo)methoxynaphthoquinones

The starting substituted 2-chloro-3,5,8-trimethoxy-1,4-naphthoquinones **5** and **6** were prepared by treatment of appropriated 2,3-dichloro-5,8-dimethoxynaphtalene-1,4-dione and 2,3,6,7-tetrachloro-5,8-dimethoxynaphtalene-1,4-dione with AcONa in dry methanol at reflux as described earlier [27] (Figure 1). Bromination of available hydroxyjuglone derivatives **7** and **8** in chloroform solution according to the work in [28] led to bromoquinones **9** and **10** in good yields, 92%. Subsequent treatment of quinones **9** and **10** in 1,4-dioxane with 0.2 M ethereal diazomethane solution and crystallization from MeOH gave a second pair of initial bromomethoxyjuglones **11** and **12** in yields 85% (Scheme 2), which were identical to such compounds described in [29].

### 2.2. Synthesis of 3-Acetylthioglycosides of 2-Methoxynaphthoquinones **13a**–**d**–**16a**–**d**

Then, four substituted 1,4-naphthoquinones—**5**, **6**, **11**, and **12**—were condensed in acetone with per-*O*-acetylated 1-thiosugars **2a**–**d** at equimolar ratio of quinone: thiosugar under base conditions in presence of K_2_CO_3_ according the procedure described in our previous work [22] (Scheme 1). This condensation led to acetylated thioglycosides of 1,4-naphthoquinones **13a**–**d**–**16a**–**d** in good yields, 71–91% (Figure 2). The structures of new compounds were proved by NMR, IR spectroscopy, and HR mass spectrometry. The 1′,2′-*trans*-configuration of glycosidic bond was confirmed by the value of anomeric proton doublets (*J_1′,2′_* = 7.5–10.2 Hz) in ^1^H-NMR spectra. The other spectral characteristics of the naphthoquinone methoxyderivatives **13a**–**d**–**16a**–**d** were in a good agreement with their proposed structures (see also Appendix A).

### 2.3. Preparation of Naphthoquinone–Sugar Tetracyclic Conjugates

Under the base treatment by MeONa/MeOH thioglycosides **13a**–**d**–**16a**–**d** were readily converted in tetracycles **17a**–**d**–**20a**–**d** in good yields 82–97% (Figure 3). It is evident that tetracyclic quinone–glucoside conjugates **17a**–**d**–**20a**–**d** were formed from methoxyglucosides **13a**–**d**–**16a**–**d** through deacetylation stage and intramolecular nucleophilic substitution of the methoxy group by sugar C_2_-OH group.

This process proceeds with retention of the configuration of all asymmetric centers of the carbohydrate portion and formation of linear tetracyclic structure. The alternative angular structure for the obtained tetracycles **17a**–**d**–**20a**–**d** was rejected based upon on the direct comparison with the spectral data of similar angular tetracycles, which we obtained earlier [30].

### 2.4. Biological Evaluation

#### 2.4.1. Cytotoxic Activity

Tetracyclic conjugates **17a**–**d**–**20a**–**d** and 5-hydroxy-1,4-naphthoquinone (juglone) were examined for cytotoxicity against three cancer cell lines and two normal cell lines such as human cervical cancer (HeLa), mouse neuroblastoma (Neuro 2a), mouse ascites Ehrlich carcinoma, mouse normal epithelial cell line (JB6 Cl 41-5a), and mouse blood erythrocytes. Known cytotoxic agent cucumarioside A_2_-2 [31] was used as positive control. The results are presented in Table 1.

Conjugates **17a**,**c**,**d** with a 7,10-dimethoxynaphthoquinone core were inactive for all cell lines at EC_50_ value ≤ 25 μM. Galactoside derivative **17b** had poor solubility in DMSO, which did not allow for the measurement its activity. Introduction of two chlorine atoms in a 7,10-dimethoxynaphthoquinone core led to tetracyclic 8,9-dichloroderivatives **18a**–**d** with better solubility and promising activity in EC_50_ values ranging from 1.1 to 10.9 μM. Among the substances of this group, galactoside derivative **18b** showed the best activity against ascites Ehrlich carcinoma cell line with EC_50_ value 1.1 μM.

The following two groups of tetracycles, **19a**–**d** and **20a**–**d**, are conjugates of thiosaccharides **2a**–**d** with derivatives of 5-hydroxy-1,4-naphthoquinone **11**–**12**. Among them, six substances had promising values, EC_50_ < 1 μM. In the group of tetracycles **19a**–**d**, bearing a hydroxyl group at position 10, only hexapyranoside derivatives d-Glc **19a** and d-Gal **19b** showed the high cytotoxic activity with EC_50_ values in the range of 0.6 to 0.9 μM, while all tetracycles **20a**–**d**, bearing hydroxyl group at position 7, revealed highly toxic compounds with EC_50_ 0.3–0.7 μM for various types of cells. Mouse epithelial Jb6 cells were more susceptible to the action of juglone tetracyclic derivatives **20a**–**d**. As evidenced from Table 1, the presence of a hydroxy group in the naphthoquinone scaffold led to the formation of naphthoquinone tetracycles **19a**–**b**–**20a**–**b** with promising cytotoxicity. However, unsubstituted 5-hydroxy-1,4-naphthoquinone (juglone) did not show any cytotoxic activity up to 100 μM. This fact proves the positive effect of heterocyclization on tetracycles cytotoxicity. Moreover, it was shown that all tested compounds did not cause lysis of murine erythrocytes up to 25 μM.

For all cytotoxic compounds **18a**–**d**–**20a**–**d** their selectivity index (SI) was calculated (Table 2). Among the tested tetracycles, the most selective substance was **19a** in relation to all studied lines of tumor cells. In comparison to non-tumor mouse epithelial Jb6 Cl 41-5a cells, the selectivity index for Ehrlich carcinoma cells was 9.3, for HeLa—1.5, and for Neuro-2a cell cuture—1.4. Compounds **18b** and **19d** also showed rather high values of the selectivity index for Ehrlich carcinoma cells with SI = 2.6 and SI = 2.1, respectively. On HeLa cells, the most active compounds were **19a**, **19d**, and **20d**, with SI > 1.

#### 2.4.2. Antimicrobial Activity

All synthesized tetracyclic conjugates **17a**–**b**–**20a**–**b** and 5-hydroxy-1,4-naphthoquinone (juglone) were screened by the agar diffusion method for antibacterial activity against two strains of Gram-positive bacteria (*S. aureus* ATCC 21027; *B. cereus* ATCC 10702), two strains of Gram-negative bacteria (*P. aeruginosa* ATCC 27853; *E. coli* K-12), and antifungal activity against fungus *C. albicans* KMM 453 from the Collection of Marine Microorganisms (KMM) of the G.B. Elyakov Pacific Institute of Bioorganic Chemistry. Commercial antibacterial drugs (vancomicin and gentamicin) and antifungal (clotrimazol) drugs were used as positive controls. Compound concentrations and diameter of inhibition zone are presented in Table 3.

Conjugates **17a**–**c** with a 7,10-dimethoxynaphthoquinone core were inactive to Gram-positive and Gram-negative bacteria in concentrations of 100 µg/well. Only one of these conjugates—**17d**, based on l-arabinose, revealed weak activity against *C. albicans*. Conjugates **18c**,**d** with two chlorine atoms in their naphthoquinone skeleton showed moderate activity against Gram-positive strains *S. aureus* and *B. cereus* at a concentration of 100 µg/well, but also were not active to Gram-negative bacteria and fungus *C. albicans*. As evidenced from Table 3, tetracycles **19b**–**d**, bearing a hydroxyl group at C-10 atom of naphthoquinone core, showed various antimicrobial activity levels to Gram-positive bacteria from weak activity for tetracycle **19b**, to strong activity for compound **19d**.

Sugar tetracylic conjugates **20a**–**d**, with a hydroxyl group at the C-7 atom of their naphthoquinone core, constituted the most effective set of antimicrobials among tested compounds **17a**–**d**–**20a**–**d**. All these compounds, **20a**–**d**, have an inhibition zone diameter of 22–40 mm at concentration 100 µg/well and kept the value of inhibition zone in the range of 10 to 25 mm for the concentration 10 µg/well. At a concentration of 10 µg/well, the most effective conjugates with d-xylose **20c** and l-arabinose **20d** showed antimicrobial activity comparable with antibiotics vancomicin and gentamicin, and these ones retained residual effect upon dilution to 1 µg/well for Gram-positive strains. Surprisingly, 5-hydroxy-1,4-naphthoquinone (juglone) showed moderate activity against only *P. aeruginosa*. Thus, heterocyclization with 1-thiosugars leads to an increase of antibiotic activity, especially against the *S. aureus* strain.

The antimicrobial activity for the most active compounds, **19d** and **20a**–**d**, was also determined against *S. aureus* by the minimum inhibitory concentration (MIC) using the broth microdilution method. As it follows from Table 4, juglone derivatives **19d** and **20d** with d-arabinose moiety showed highest antibacterial activity with a MIC of 6.25 μM. The worse activity for **19d** in the agar medium test is probably associated with interactions with agar molecules in the medium. In comparison to non-tumor mouse epithelial Jb6 Cl 41-5a cells, no selective inhibitory activity on bacterial cells was observed, as SI < 1.

## 3. Materials and Methods

### 3.1. General Information

All reagents were obtained from commercial suppliers and were used without additional purification. All solvents were distilled before use. Melting points were determined by using a Boetius apparatus (VEB Analytic, Dresden, Germany) and are uncorrected. IR spectra were recorded in KBr pellets or in CHCl_3_ by using a Bruker Equinox 55 spectrophotometer (Bruker Optik GmbH, Ettlingen, Germany). ^1^H-NMR spectra were recorded on a Bruker Avance III-500 HD (500 MHz) or a Bruker Avance III-700 (700 MHz) spectrometer (Bruker Corporation, Bremen, Germany) with CDCl_3_ or DMSO-*d*_6_ as the solvent and TMS as the internal standard. ^13^C-NMR spectra were recorded on a Bruker Avance III HD-500 spectrometer at 125 MHz or a Bruker Avance III-700 spectrometer at 176 MHz. ESI mass spectra were recorded on an Agilent 6510 Q-TOF LC/MS instrument (Agilent, Santa Clara, CA, USA). Silufol UV–Vis TLC plates (Sklarny Kavalier, Votitsa, Czech Republic) treated with HCl vapor were used for analytical TLC. Preparative TLC was performed on silica gel 60 Merck (40–60 μm). TLC plates were developed in system A (hexane–benzene–acetone (2:1:1 *v*/*v*)), system B (hexane–benzene–acetone (2:1:2 *v*/*v*)), or system C (benzene–EtOAc–MeOH (7:4:2 *v*/*v*)).

### 3.2. Synthesis

#### 3.2.1. General Procedure for Synthesis of Acetylated Thioglycosides **13a–d–16a**–**d**

2-Chloro(bromo)-3-methoxyquinone, **5**, **6**, **11**, and **12** (0.50 mM), and per-*O*-acetylated 1-mercaptosugar derivatives **2a**–**d** (0.55 mM) were dissolved in acetone (30 mL) and 76 mg (0.55 mM) of dry finely powdered K_2_CO_3_ was added. The resulting mixture was stirred for 2 h at room temperature until the consumption of thioglicose and conversion of starting quinone. Precipitate of inorganic salt was filtered, the filtrate was dried in a vacuum, and the residue was subjected to preparative TLC (system B for **13a**–**d** and system A for others). The main fraction was washed off from silica gel with acetone, dried, and recrystallized from MeOH to give pure thioglycoside **13a**–**d**–**16a**–**d**.

*2-(2,3,4,6-Tetra-O-acetyl-β-d-glucopyranosyl-1-thio)-3,5,8-trimethoxy-1,4-naphthoquinone* (**13a**); yield 276 mg (90%), orange solid, *R_f_* 0.37 (B), m.p. 162–165 °C. ^1^H-NMR (500 MHz, CDCl_3_), δ: 1.97 (s, 3H, COCH_3_), 2.00 (s, 3H, COCH_3_), 2.01 (s, 3H, COCH_3_), 2.06 (s, 3H, COCH_3_), 3.70 (m, 1H, H-5′), 3.93 (s, 3H, -OCH_3_), 3.94 (s, 3H, -OCH_3_), 3.99 (dd, 1H, *J* = 12.4, 1.9 Hz, H-6′a), 4.14 (s, 3H, -OCH_3_), 4.20 (dd, 1H, *J* = 12.4, 4.5, Hz, H-6′b), 5.10 (t, 1H, *J* = 9.6 Hz, H-2′), 5.12 (t, 1H, *J* = 9.7 Hz, H-4′), 5.26 (t, 1H, *J* = 9.2 Hz, H-3′), 5.61 (d, 1H, *J* = 10.1 Hz, H-1′), 7.23 (d, 1H, *J* = 9.5 Hz, Ar-H), 7.27 (d, 1H, *J* = 9.5 Hz, Ar-H). ^13^C-NMR(125 MHz, CDCl_3_), δ: 20.6 (3 × COCH_3_), 20.7 (COCH_3_), 56.9 (-OCH_3_), 57.1 (-OCH_3_), 61.1 (-OCH_3_), 61.8 (C-6′), 68.2 (C-4′), 71.2 (C-2′), 74.1 (C-3′), 75.7 (C-5′), 80.7 (C-1′), 119.4, 120.5, 120.7, 122.2, 126.9 (C-2), 153.0, 153.7, 158.3 (C-3), 169.3 (COCH_3_), 169.4 (COCH_3_), 170.2 (COCH_3_), 170.6 (COCH_3_), 178.0, 181.4. IR (CHCl_3_): 3050, 2943, 2842, 1755, 1663, 1597, 1571, 1479, 1463, 1435, 1413, 1368, 1334, 1270, 1243, 1210, 1193 cm^−1^. HRMS (ESI): *m*/*z* [M − H]^−^ calcd. for C_27_H_29_O_14_S 609.1284, found 609.1281.

*2-(2,3,4,6-Tetra-O-acetyl-β-d-galactopyranosyl-1-thio)-3,5,8-trimethoxy-1,4-naphthoquinone* (**13b**); yield 253 mg (82.5%), orange solid, *R_f_* 0.37 (B), m.p. 99–101 °C. ^1^H-NMR (500 MHz, CDCl_3_), δ: 1.94 (s, 3H, COCH_3_), 1.98 (s, 3H, COCH_3_), 2.08 (s, 3H, COCH_3_), 2.13 (s, 3H, COCH_3_), 3.91 (m, 1H, H-5′), 3.93 (s, 3H, -OCH_3_), 3.95 (s, 3H, -OCH_3_), 4.02 (m, 2H, H-6′a, H-6′b), 4.14 (s, 3H, -OCH_3_), 5.10 (dd, 1H, *J* = 9.9, 3.5 Hz, H-3′), 5.31 (t, 1H, *J* = 10.0 Hz, H-2′), 5.42 (m, 1H, H-4′), 5.59 (d, 1H, *J* = 10.2 Hz, H-1′), 7.23 (d, 1H, *J* = 9.5 Hz, Ar-H), 7.27 (d, 1H, *J* = 9.5 Hz, Ar-H). ^13^C-NMR(125 MHz, CDCl_3_), δ: 20.5 (COCH_3_), 20.6 (2 × COCH_3_), 20.8 (COCH_3_), 56.9 (-OCH_3_), 57.0 (-OCH_3_), 61.0 (C-6′), 61.1 (-OCH_3_), 67.2 (C-4′), 68.4 (C-2′), 72.0 (C-3′), 74.3 (C-5′), 81.5 (C-1′), 119.4, 120.5, 120.6, 122.1, 126.9 (C-2), 153.1, 153.7, 158.5 (C-3), 169.6 (COCH_3_), 170.0 (2 × COCH_3_), 170.2 (COCH_3_), 178.1, 181.3. IR (CHCl_3_): 3054, 3006, 2941, 2842, 1750, 1663, 1597, 1571, 1479, 1463, 1435, 1413, 1372, 1334, 1270, 1251, 1185, 1155, 1085, 1060, 1022 cm^−1^. HRMS (ESI): *m*/*z* [M − H]^−^ calcd. for C_27_H_29_O_14_S 609.1284, found 609.1283.

*2-(2,3,4-Tri-O-acetyl-β-d-xylopyranosyl-1-thio)-3,5,8-trimethoxy-1,4-naphthoquinone* (**13c**); yield 230 mg (85%), orange solid, *R_f_* 0.39 (B), m.p. 197–199 °C. ^1^H-NMR (500 MHz, CDCl_3_), δ: 2.04 (s, 3H, COCH_3_), 2.06 (s, 3H, COCH_3_), 2.08 (s, 3H, COCH_3_), 3.39 (dd, 1H, *J* = 11.9, 8.3 Hz, H-5′a), 3.93 (s, 3H, -OCH_3_), 3.95 (s, 3H, -OCH_3_), 4.15 (s, 3H, -OCH_3_), 4.21 (dd, 1H, *J* = 11.9, 4.7 Hz, H-5′b), 4.94 (m, 1H, H-4′), 5.04 (t, 1H, *J* = 8.0 Hz, H-2′), 5.20 (t, 1H, *J* = 8.0 Hz, H-3′), 5.60 (d, 1H, *J* = 8.0 Hz, H-1′), 7.24 (d, 1H, *J* = 9.5 Hz, Ar-H), 7.27 (d, 1H, *J* = 9.5 Hz, Ar-H). ^13^C-NMR(125 MHz, CDCl_3_), δ: 20.7 (3 × COCH_3_), 56.9 (-OCH_3_), 57.1 (-OCH_3_), 61.2 (-OCH_3_), 64.9 (C-5′), 68.6 (C-4′), 70.6 (C-2′), 71.7 (C-3′), 81.6 (C-1′), 119.4, 120.6, 120.8, 122.0, 126.6 (C-2), 153.3, 153.8, 159.4 (C-3), 169.4 (COCH_3_), 169.7 (COCH_3_), 169.8 (COCH_3_), 178.3, 181.3. IR (CDCl_3_): 3054, 3018, 3006, 2942, 2842, 1752, 1663, 1597, 1571, 1478, 1463, 1435, 1413, 1371, 1334, 1272, 1248, 1210, 1185, 1062, 1024 cm^−1^. HRMS (ESI): *m*/*z* [M − H]^−^ calcd. for C_24_H_25_O_12_S 537.1072, found 537.1073.

*2-(2,3,4-Tri-O-acetyl-α-l-arabinopyranosyl-1-thio)-3,5,8-trimethoxy-1,4-naphthoquinone* (**13d**); yield 236 mg (87.2%), dark orange solid, *R_f_* 0.39 (B), m.p. 92–94 °C. ^1^H-NMR (500 MHz, CDCl_3_), δ: 2.07 (s, 3H, COCH_3_), 2.10 (s, 3H, COCH_3_), 2.11 (s, 3H, COCH_3_), 3.64 (dd, 1H, *J* = 12.6, 2.3 Hz, H-5′a), 3.92 (s, 3H, -OCH_3_), 3.94 (s, 3H, -OCH_3_), 4.08 (dd, 1H, *J* = 12.6, 4.3 Hz, H-5′b), 4.15 (s, 3H, -OCH_3_), 5.14 (dd, 1H, *J* = 8.2, 3.4 Hz, H-3′), 5.27 (m, 1H, H-4′), 5.31 (t, 1H, *J* = 7.9 Hz, H-2′), 5.60 (d, 1H, *J* = 7.9 Hz, H-1′), 7.23 (d, 1H, *J* = 9.5 Hz, Ar-H), 7.26 (d, 1H, *J* = 9.5 Hz, Ar-H). ^13^C-NMR(125 MHz, CDCl_3_), δ: 20.7 (COCH_3_), 20.8 (COCH_3_), 20.9 (COCH_3_), 56.9 (-OCH_3_), 57.2 (-OCH_3_), 61.2 (-OCH_3_), 65.3 (C-5′), 67.6 (C-4′), 69.2 (C-2′), 70.3 (C-3′), 81.8 (C-1′), 119.4, 120.6, 120.7, 122.1, 127.1 (C-2), 153.2, 153.7, 159.1 (C-3), 169.5 (COCH_3_), 169.8 (COCH_3_), 170.2 (COCH_3_), 178.3, 181.4. IR (CHCl_3_): 3054, 3005, 2941, 2841, 1747, 1661, 1597, 1570, 1478, 1463, 1435, 1412, 1372, 1335, 1272, 1250, 1185, 1159, 1105, 1087, 1060, 1022 cm^−1^. HRMS (ESI): *m*/*z* [M − H]^−^ calcd. for C_24_H_25_O_12_S 537.1072, found 537.1070.

*2-(2,3,4,6-Tetra-O-acetyl-β-d-glucopyranosyl-1-thio)-6,7-dichloro-3,5,8-trimethoxy-1,4-naphthoquinone* (**14a**); yield 279 mg (82%), yellow solid, *R_f_* 0.48 (A), m.p. 103–105 °C. ^1^H-NMR (500 MHz, CDCl_3_), δ: 1.96 (s, 3H, COCH_3_), 2.01 (s, 3H, COCH_3_), 2.02 (s, 3H, COCH_3_), 2.08 (s, 3H, COCH_3_), 3.67 (m, 1H, H-5′), 3.98 (s, 6H, 2 × -OCH_3_), 4.01 (dd, 1H, *J* = 12.6, 2.2 Hz, H-6′a), 4.15 (s, 3H, -OCH_3_), 4.18 (dd, 1H, *J* = 12.6, 4.7 Hz, H-6′b), 5.10 (m, 2H, H-2′, H-4′), 5.27 (t, 1H, *J* = 9.3 Hz, H-3′), 5.60 (d, 1H, *J* = 10.2 Hz, H-1′). ^13^C-NMR(125 MHz, CDCl_3_), δ: 20.5 (2 × COCH_3_), 20.6 (2 × COCH_3_), 61.3 (-OCH_3_), 61.8 (C-6′), 62.3 (-OCH_3_), 62.4 (-OCH_3_), 68.1 (C-4′), 71.1 (C-2′), 74.0 (C-3′), 76.0 (C-5′), 80.4 (C-1′), 124.1, 125.7, 127.6 (C-2), 136.2, 136.9, 152.6, 153.0, 158.3 (C-3), 169.3 (COCH_3_), 169.4 (COCH_3_), 170.1 (COCH_3_), 170.5 (COCH_3_), 176.4, 179.7. IR (CHCl_3_): 3050, 2944, 2857, 1756, 1670, 1581, 1547, 1459, 1440, 1380, 1326, 1304, 1241, 1207, 1194, 1030 cm^−1^. HRMS (ESI): *m*/*z* [M − H]^−^ calcd. for C_27_H_27_Cl_2_O_14_S 677.0504, found 677.0504.

*2-(2,3,4,6-Tetra-O-acetyl-β-d-galactopyranosyl-1-thio)-6,7-dichloro-3,5,8-trimethoxy-1,4-naphthoquinone* (**14b**); yield 302 mg (88.8%), orange solid, *R_f_* 0.48 (A), m.p. 80–81 °C. ^1^H-NMR (500 MHz, CDCl_3_), δ: 1.95 (s, 3H, COCH_3_), 1.99 (s, 3H, COCH_3_), 2.09 (s, 3H, COCH_3_), 2.14 (s, 3H, COCH_3_), 3.90 (m, 1H, H-5′), 3.98 (s, 6H, 2 × -OCH_3_), 4.03 (m, 2H, H-6′a, H-6′b), 4.16 (s, 3H, -OCH_3_), 5.11 (dd, 1H, *J* = 9.9, 3.5 Hz, H-3′), 5.30 (t, 1H, *J* = 10.0 Hz, H-2′), 5.42 (m, 1H, H-4′), 5.57 (d, 1H, *J* = 10.2 Hz, H-1′). ^13^C-NMR(125 MHz, CDCl_3_), δ: 20.5 (2 × COCH_3_), 20.6(COCH_3_), 20.7 (COCH_3_), 61.2 (C-6′), 61.7 (-OCH_3_), 62.3 (-OCH_3_), 62.4 (-OCH_3_), 67.2 (C-4′), 68.3 (C-2′), 71.9 (C-3′), 74.6 (C-5′), 81.3 (C-1′), 124.1, 125.6, 127.7 (C-2), 136.2, 136.9, 152.7, 153.0, 158.4 (C-3), 169.6 (COCH_3_), 170.0 (COCH_3_), 170.2 (COCH_3_), 170.3 (COCH_3_), 176.5, 179.7. IR (CHCl_3_): 3053, 3007, 2944, 2856, 1751, 1670, 1580, 1459, 1440, 1380, 1326, 1304, 1245, 1191, 1113, 1086, 1059, 1028 cm^−1^. HRMS (ESI): *m*/*z* [M − H]^−^ calcd. for C_27_H_27_Cl_2_O_14_S 677.0504, found 677.0501.

*2-(2,3,4-Tri-O-acetyl-β-d-xylopyranosyl-1-thio)-6,7-dichloro-3,5,8-trimethoxy-1,4-naphthoquinone* (**14c**); yield 263 mg (86.5%), dark orange solid, *R_f_* 0.52 (A), m.p. 132–134 °C. ^1^H-NMR (500 MHz, CDCl_3_), δ: 2.04 (s, 3H, COCH_3_), 2.07 (s, 3H, COCH_3_), 2.10 (s, 3H, COCH_3_), 3.36 (dd, 1H, *J* = 11.8, 8.6 Hz, H-5′a), 3.97 (s, 3H, -OCH_3_), 3.98 (s, 3H, -OCH_3_), 4.15 (s, 3H, -OCH_3_), 4.18 (dd, 1H, *J* = 11.8, 5.0 Hz, H-5′b), 4.95 (m, 1H, H-4′), 5.04 (t, 1H, *J* = 8.2 Hz, H-2′), 5.22 (t, 1H, *J* = 8.2 Hz, H-3′), 5.57 (d, 1H, *J* = 8.2 Hz, H-1′). ^13^C-NMR(125 MHz, CDCl_3_), δ: 20.7 (3 × COCH_3_), 61.3 (-OCH_3_), 62.3 (2×-OCH_3_), 65.2 (C-5′), 68.5 (C-4′), 70.7 (C-2′), 71.9 (C-3′), 81.3 (C-1′), 124.1, 125.4, 127.6 (C-2), 136.2, 136.9, 152.8, 153.0, 159.1 (C-3), 169.4 (COCH_3_), 169.7 (COCH_3_), 169.9 (COCH_3_), 176.7, 179.7. IR (CHCl_3_): 3054, 3019, 3006, 2944, 2857, 1754, 1671, 1580, 1547, 1459, 1440, 1380, 1326, 1304, 1246, 1190, 1114, 1067, 1029 cm^−1^. HRMS (ESI): *m*/*z* [M − H]^−^ calcd. for C_24_H_23_Cl_2_O_12_S 605.0293, found 605.0291.

*2-(2,3,4-Tri-O-acetyl-α-l-arabinopyranosyl-1-thio)-6,7-dichloro-3,5,8-trimethoxy-1,4-naphthoquinone* (**14d**); yield 276 mg (90.8%), dark orange solid, *R_f_* 0.50 (A), m.p. 79–81 °C. ^1^H-NMR (500 MHz, CDCl_3_), δ: 2.07 (s, 3H, COCH_3_), 2.11 (s, 6H, 2 × COCH_3_), 3.63 (dd, 1H, *J* = 12.7, 1.9 Hz, H-5′a), 3.96 (s, 3H, -OCH_3_), 3.98 (s, 3H, -OCH_3_), 4.06 (dd, 1H, *J* = 12.7, 4.2 Hz, H-5′b), 4.16 (s, 3H, -OCH_3_), 5.16 (dd, 1H, *J* = 8.3, 3.4 Hz, H-3′), 5.28 (m, 1H, H-4′), 5.31 (t, 1H, *J* = 8.0 Hz, H-2′), 5.57 (d, 1H, *J* = 8.0 Hz, H-1′). ^13^C-NMR(125 MHz, CDCl_3_), δ: 20.7 (COCH_3_), 20.8 (COCH_3_), 20.8 (COCH_3_), 61.3 (-OCH_3_), 62.3 (2 × -OCH_3_), 65.6 (C-5′), 67.5 (C-4′), 69.2 (C-2′), 70.3 (C-3′), 81.6 (C-1′), 124.1, 125.5, 128.0 (C-2), 136.1, 136.9, 152.8, 153.0, 158.9 (C-3), 169.4 (COCH_3_), 169.9 (COCH_3_), 170.2 (COCH_3_), 176.7, 179.8. IR (CHCl_3_): 3054, 3027, 3005, 2943, 2856, 1748, 1670, 1580, 1546, 1459, 1440, 1380, 1326, 1304, 1247, 1225, 1190, 1113, 1087, 1060, 1028 cm^−1^. HRMS (ESI): *m*/*z* [M − H]^−^ calcd. for C_24_H_23_Cl_2_O_12_S 605.0293, found 605.0289.

*2-(2,3,4,6-Tetra-O-acetyl-β-d-glucopyranosyl-1-thio)-8-hydroxy-3-methoxy-1,4-naphthoquinone* (**15a**); yield 210 mg (73.9%), orange solid, *R_f_* 0.46 (A), m.p. 158–161 °C. ^1^H-NMR (700 MHz, CDCl_3_): δ 1.93 (s, 3H, COCH_3_), 2.02 (s, 3H, COCH_3_), 2.03 (s, 3H, COCH_3_), 2.09 (s, 3H, COCH_3_), 3.72 (m, 1H, H-5′), 4.08 (dd, 1H, *J* = 12.4, 2.0 Hz, H-6′a), 4.15 (dd, 1H, *J* = 12.4, 5.5 Hz, H-6′b), 4.29 (s, 3H, -OCH_3_), 5.08 (t, 1H, *J* = 9.7 Hz, H-4′), 5.11 (t, 1H, *J* = 10.1 Hz, H-2′), 5.27 (t, 1H, *J* = 9.3 Hz, H-3′), 5.50 (d, 1H, *J* = 10.1 Hz, H-1′), 7.26 (dd, 1H, *J* = 7.4, 1.2 Hz, H-7), 7.59 (dd, 1H, *J* = 8.0, 7.4 Hz, H-6), 7.61 (dd, 1H, *J* = 8.0, 1.2 Hz, H-5), 12.03 (s, 1H, C^8^OH). ^13^C-NMR(176 MHz, CDCl_3_): δ 20.4 (COCH_3_), 20.5 (COCH_3_), 20.6 (COCH_3_), 20.7 (COCH_3_), 62.1 (C-6′, -OCH_3_), 68.3 (C-4′), 71.3 (C-2′), 73.9 (C-3′), 76.0 (C-5′), 81.7 (C-1′), 114.3 (C-9), 119.7 (C-5), 125.0 (C-7), 126.0 (C-2), 131.3 (C-10), 135.8 (C-6), 160.5 (C-3), 161.2 (C-8), 169.3 (COCH_3_), 169.4 (COCH_3_), 170.2 (COCH_3_), 170.5 (COCH_3_), 178.4 (C-4), 187.7 (C-1). IR (CHCl_3_): 3053, 3007, 2953, 1756, 1669, 1630, 1580, 1559, 1458, 1369, 1313, 1250, 1191, 1162, 1077, 1050 cm^−1^. HRMS (ESI): *m*/*z* [M − H]^−^ calcd. for C_25_H_25_O_13_S 565.1021, found 565.1016.

*2-(2,3,4,6-Tetra-O-acetyl-β-d-galactopyranosyl-1-thio)-8-hydroxy-3-methoxy-1,4-naphthoquinone* (**15b**); yield 203 mg (71.4%), orange solid, *R_f_* 0.46 (A), m.p. 190–193 °C. ^1^H-NMR (500 MHz, CDCl_3_), δ: 1.91 (s, 3H, COCH_3_), 1.99 (s, 3H, COCH_3_), 2.10 (s, 3H, COCH_3_), 2.15 (s, 3H, COCH_3_), 3.92 (m, 1H, H-5′), 4.06 (m, 2H, H-6′a, H-6′b), 4.30 (s, 3H, -OCH_3_), 5.10 (dd, 1H, *J* = 9.9, 3.5 Hz, H-3′), 5.32 (t, 1H, *J* = 10.0 Hz, H-2′), 5.43 (m, 1H, H-4′), 5.46 (d, 1H, *J* = 10.0 Hz, H-1′), 7.26 (dd, 1H, *J* = 7.7, 1.5 Hz, H-7), 7.58 (dd, 1H, *J* = 7.7, 7.5 Hz, H-6), 7.61 (dd, 1H, *J* = 7.5, 1.5 Hz, H-5), 12.04 (s, 1H, C^8^OH). ^13^C-NMR(125 MHz, CDCl_3_), δ: 20.4 (COCH_3_), 20.5 (COCH_3_), 20.6 (COCH_3_), 20.8 (COCH_3_), 61.6 (C-6′), 62.1 (-OCH_3_), 67.3 (C-4′), 68.4 (C-2′), 71.9 (C-3′), 74.8 (C-5′), 82.5 (C-1′), 114.3 (C-9), 119.6 (C-5), 125.0 (C-7), 126.2 (C-2), 131.4 (C-10), 135.8 (C-6), 160.5 (C-3), 161.3 (C-8), 169.6 (COCH_3_), 170.0 (COCH_3_), 170.2 (COCH_3_), 170.3 (COCH_3_), 178.5 (C-4), 187.7 (C-1). IR (CHCl_3_): 3055, 3019, 2954, 1751, 1669, 1630, 1580, 1559, 1458, 1441, 1370, 1312, 1249, 1193, 1162, 1080, 1053 cm^−1^. HRMS (ESI): *m*/*z* [M − H]^−^ calcd. for C_25_H_25_O_13_S 565.1021, found 565.1021.

*2-(2,3,4-Tri-O-acetyl-β-d-xylopyranosyl-1-thio)-8-hydroxy-3-methoxy-1,4-naphthoquinone* (**15c**); yield 205 mg (82.6%), orange solid, *R_f_* 0.50 (A), m.p. 133–135 °C. ^1^H-NMR (500 MHz, CDCl_3_), δ: 2.04 (s, 3H, COCH_3_), 2.08 (s, 3H, COCH_3_), 2.11 (s, 3H, COCH_3_), 3.41 (dd, 1H, *J* = 11.9, 8.1 Hz, H-5′a), 4.25 (dd, 1H, *J* = 11.9, 4.8 Hz, H-5′b), 4.29 (s, 3H, -OCH_3_), 4.95 (m, 1H, H-4′), 5.05 (t, 1H, *J* = 7.8 Hz, H-2′), 5.21 (t, 1H, *J* = 7.8 Hz, H-3′), 5.51 (d, 1H, *J* = 7.9 Hz, H-1′), 7.26 (dd, 1H, *J* = 8.0, 1.7 Hz, H-7), 7.57 (dd, 1H, *J* = 8.0, 7.5 Hz, H-6), 7.60 (dd, 1H, *J* = 7.5, 1.5 Hz, H-5), 12.05 (s, 1H, C^8^OH). ^13^C-NMR(125 MHz, CDCl_3_), δ: 20.7 (3 × COCH_3_), 62.1 (-OCH_3_), 64.9 (C-5′), 68.3 (C-4′), 70.6 (C-2′), 71.5 (C-3′), 82.7 (C-1′), 114.4 (C-9), 119.6 (C-5), 125.0 (C-7), 126.0 (C-2), 131.4 (C-10), 135.8 (C-6), 161.3 (C-8), 161.4 (C-3), 169.4 (COCH_3_), 169.7 (COCH_3_), 169.8 (COCH_3_), 178.8 (C-4), 187.7 (C-1). IR (CHCl_3_): 3056, 2953, 1752, 1671, 1630, 1580, 1558, 1458, 1371, 1313, 1249, 1240, 1208, 1163, 1076 cm^−1^. HRMS (ESI): *m*/*z* [M − H]^−^ calcd. for C_22_H_21_O_11_S 493.0810, found 493.0805.

*2-(2,3,4-Tri-O-acetyl-α-l-arabinopyranosyl-1-thio)-8-hydroxy-3-methoxy-1,4-naphthoquinone* (**15d**); yield 197 mg (79.4%), orange solid, *R_f_* 0.50 (A), m.p. 84–86 °C. ^1^H-NMR (500 MHz, CDCl_3_), δ: 2.10 (s, 3H, COCH_3_), 2.12 (s, 6H, 2 × COCH_3_), 3.65 (dd, 1H, *J* = 12.5, 2.5 Hz, H-5′a), 4.14 (dd, 1H, *J* = 12.5, 4.9 Hz, H-5′b), 4.30 (s, 3H, -OCH_3_), 5.17 (dd, 1H, *J* = 7.9, 3.4 Hz, H-3′), 5.28 (m, 1H, H-4′), 5.32 (t, 1H, *J* = 7.5 Hz, H-2′), 5.48 (d, 1H, *J* = 7.5 Hz, H-1′), 7.26 (dd, 1H, *J* = 7.8, 1.6 Hz, H-7), 7.57 (dd, 1H, *J* = 7.8, 7.5 Hz, H-6), 7.60 (dd, 1H, *J* = 7.5, 1.6 Hz, H-5), 12.06 (s, 1H, C^8^OH). ^13^C-NMR(125 MHz, CDCl_3_), δ: 20.7 (COCH_3_), 20.8 (COCH_3_), 20.9 (COCH_3_), 62.1 (-OCH_3_), 64.7 (C-5′), 67.2 (C-4′), 69.4 (C-2′), 70.0 (C-3′), 82.9 (C-1′), 114.4 (C-9), 119.6 (C-5), 125.0 (C-7), 126.5 (C-2), 131.3 (C-10), 135.8 (C-6), 161.2 (C-3), 161.3 (C-8), 169.4 (COCH_3_), 169.8 (COCH_3_), 170.1 (COCH_3_), 178.8 (C-4), 187.7 (C-1). IR (CHCl_3_): 3054, 3006, 1748, 1671, 1629, 1580, 1558, 1458, 1372, 1313, 1250, 1162, 1106, 1078, 1061 cm^−1^. HRMS (ESI): *m*/*z* [M − H]^−^ calcd. for C_22_H_21_O_11_S 493.0810, found 493.0808.

*2-(2,3,4,6-Tetra-O-acetyl-β-d-glucopyranosyl-1-thio)-5-hydroxy-3-methoxy-1,4-naphthoquinone* (**16a**); yield 248 mg (87.3%), red solid, *R_f_* 0.46 (A), m.p. 159–161 °C. ^1^H-NMR (500 MHz, CDCl_3_), δ: 1.93 (s, 3H, COCH_3_), 2.01 (s, 3H, COCH_3_), 2.02 (s, 3H, COCH_3_), 2.08 (s, 3H, COCH_3_), 3.73 (m, 1H, H-5′), 4.06 (dd, 1H, *J* = 12.4, 2.3 Hz, H-6′a), 4.16 (dd, 1H, *J* = 12.4, 5.2 Hz, H-6′b), 4.22 (s, 3H, -OCH_3_), 5.09 (dd, 1H, *J* = 10.1, 4.2 Hz, H-4′), 5.11 (dd, 1H, *J* = 10.1, 4.1 Hz, H-2′), 5.27 (t, 1H, *J* = 9.3 Hz, H-3′), 5.67 (d, 1H, *J* = 10.1 Hz, H-1′), 7.24 (dd, 1H, *J* = 8.1, 1.4 Hz, H-6), 7.59 (dd, 1H, *J* = 8.1, 7.5 Hz, H-7), 7.62 (dd, 1H, *J* = 7.5, 1.4 Hz, H-8), 11.72 (s, 1H, C^5^OH). ^13^C-NMR(125 MHz, CDCl_3_), δ: 20.5 (2 × COCH_3_), 20.6 (2 × COCH_3_), 62.0 (-OCH_3,_ C-6′), 68.3 (C-4′), 71.2 (C-2′), 74.0 (C-3′), 75.9 (C-5′), 81.1 (C-1′), 114.1 (C-10), 119.7 (C-8), 124.4 (C-6), 130.1 (C-2), 132.3 (C-9), 136.5 (C-7), 158.6 (C-3), 161.8 (C-5), 169.3 (COCH_3_), 169.4 (COCH_3_), 170.1 (COCH_3_), 170.5 (COCH_3_), 181.3 (C-1), 183.5 (C-4). IR (CHCl_3_): 3050, 3004, 2950, 1756, 1661, 1636, 1579, 1560, 1458, 1369, 1240, 1228, 1212, 1192, 1046 cm^−1^. HRMS (ESI): *m*/*z* [M − H]^−^ calcd. for C_25_H_25_O_13_S 565.1021, found 565.1017.

*2-(2,3,4,6-Tetra-O-acetyl-β-d-galactopyranosyl-1-thio)-5-hydroxy-3-methoxy-1,4-naphthoquinone* (**16b**); yield 233 mg (82.0%), dark orange solid, *R_f_* 0.46 (A), m.p. 83–86 °C. ^1^H-NMR (500 MHz, CDCl_3_), δ: 1.91 (s, 3H, COCH_3_), 1.99 (s, 3H, COCH_3_), 2.09 (s, 3H, COCH_3_), 2.15 (s, 3H, COCH_3_), 3.93 (m, 1H, H-5′), 4.05 (m, 2H, H-6′a, H-6′b), 4.23 (s, 3H, -OCH_3_), 5.11 (dd, 1H, *J* = 9.9, 3.4 Hz, H-3′), 5.31 (t, 1H, *J* = 10.0 Hz, H-2′), 5.43 (m, 1H, H-4′), 5.64 (d, 1H, *J* = 10.2 Hz, H-1′), 7.24 (dd, 1H, *J* = 8.2, 1.3 Hz, H-6), 7.59 (dd, 1H, *J* = 8.2, 7.5 Hz, H-7), 7.63 (dd, 1H, *J* = 7.5, 1.3 Hz, H-8), 11.72 (s, 1H, C^5^OH). ^13^C-NMR(125 MHz, CDCl_3_), δ: 20.5 (COCH_3_), 20.55 (COCH_3′_), 20.6 (COCH_3_), 20.7 (COCH_3_), 61.4 (C-6′), 62.0 (-OCH_3_), 67.3 (C-4′), 68.3 (C-2′), 71.9 (C-3′), 74.6 (C-5′), 81.9 (C-1′), 114.1 (C-10), 119.7 (C-8), 124.3 (C-6), 130.2 (C-2), 132.3 (C-9), 136.5 (C-7), 158.6 (C-3), 161.8 (C-5), 169.6 (COCH_3_), 170.0 (COCH_3_), 170.2 (COCH_3_), 170.3 (COCH_3_), 181.2 (C-1), 183.5 (C-4). IR (CHCl_3_): 3056, 3006, 1751, 1661, 1636, 1579, 1563, 1458, 1371, 1255, 1191, 1171, 1154, 1085, 1049 cm^−1^. HRMS (ESI): *m*/*z* [M − H]^−^ calcd. for C_25_H_25_O_13_S 565.1021, found 565.1019.

*2-(2,3,4-Tri-O-acetyl-β-d-xylopyranosyl-1-thio)-5-hydroxy-3-methoxy-1,4-naphthoquinone* (**16c)**; yield 181 mg (73.0%), orange solid, *R_f_* 0.50 (A), m.p. 135–137 °C. ^1^H-NMR (500 MHz, CDCl_3_), δ: 2.04 (s, 3H, COCH_3_), 2.07 (s, 3H, COCH_3_), 2.10 (s, 3H, COCH_3_), 3.40 (dd, 1H, *J* = 11.8, 8.4 Hz, H-5′a), 4.20 (dd, 1H, *J* = 11.8, 5.0 Hz, H-5′b), 4.22 (s, 3H, -OCH_3_), 4.96 (m, 1H, H-4′), 5.05 (t, 1H, *J* = 8.2 Hz, H-2′), 5.22 (t, 1H, *J* = 8.1 Hz, H-3′), 5.67 (d, 1H, *J* = 8.2 Hz, H-1′), 7.23 (dd, 1H, *J* = 8.1, 1.3 Hz, H-6), 7.59 (dd, 1H, *J* = 8.1, 7.5 Hz, H-7), 7.63 (dd, 1H, *J* = 7.5, 1.3 Hz, H-8), 11.71 (s, 1H, C^5^OH). ^13^C-NMR(125 MHz, CDCl_3_), δ: 20.7 (3 × COCH_3_), 62.1 (-OCH_3_), 65.1 (C-5′), 68.5 (C-4′), 70.6 (C-2′), 71.8 (C-3′), 82.0 (C-1′), 114.2 (C-10), 119.8 (C-8), 124.3 (C-6), 130.0 (C-2), 132.3 (C-9), 136.6 (C-7), 159.4 (C-3), 161.8 (C-5), 169.4 (COCH_3_), 169.7 (COCH_3_), 169.8 (COCH_3_), 181.3 (C-1), 183.7 (C-4). IR (CHCl_3_): 3053, 3007, 2949, 1754, 1661, 1637, 1579, 1563, 1458, 1371, 1248, 1192, 1171, 1070, 1048 cm^−1^. HRMS (ESI): *m*/*z* [M − H]^−^ calcd. for C_22_H_21_O_11_S 493.0810, found 493.0811.

*2-(2,3,4-Tri-O-acetyl-α-l-arabinopyranosyl-1-thio)-5-hydroxy-3-methoxy-1,4-naphthoquinone* (**16d**); yield 189 mg (76.2%), dark orange solid, *R_f_* 0.50 (A), m.p. 97–99 °C. ^1^H-NMR(700 MHz, CDCl_3_), δ: 2.08 (s, 3H, COCH_3_), 2.11 (s, 3H, COCH_3_), 2.12 (s, 3H, COCH_3_), 3.66 (dd, 1H, *J* = 12.6, 2.3 Hz, H-5′a), 4.09 (dd, 1H, *J* = 12.6, 4.4 Hz, H-5′b), 4.23 (s, 3H, -OCH_3_), 5.16 (dd, 1H, *J* = 8.2, 3.4 Hz, H-3′), 5.29 (m, 1H, H-4′), 5.31 (t, 1H, *J* = 7.7 Hz, H-2′), 5.66 (d, 1H, *J* = 7.7 Hz, H-1′), 7.23 (dd, 1H, *J* = 8.3, 1.0 Hz, H-6), 7.59 (dd, 1H, *J* = 8.3, 7.5 Hz, H-7), 7.63 (dd, 1H, *J* = 7.5, 1.0 Hz, H-8), 11.72 (s, 1H, C^5^OH). ^13^C-NMR (176 MHz, CDCl_3_), δ: 20.7 (COCH_3_), 20.8 (COCH_3_), 20.9 (COCH_3_), 62.0 (-OCH_3_), 65.3 (C-5′), 67.4 (C-4′), 69.2 (C-2′), 70.2 (C-3′), 82.2 (C-1′), 114.2 (C-10), 119.8 (C-8), 124.3 (C-6), 130.5 (C-2), 132.3 (C-9), 136.6 (C-7), 159.3 (C-3), 161.8 (C-5), 169.5 (COCH_3_), 169.8 (COCH_3_), 170.2 (COCH_3_), 181.4 (C-1), 183.8 (C-4). IR (CHCl_3_): 3056, 2947, 1748, 1661, 1636, 1579, 1562, 1457, 1371, 1240, 1193, 1170, 1105, 1087, 1066, 1024 cm^−1^. HRMS (ESI): *m*/*z* [M − H]^−^ calcd. for C_22_H_21_O_11_S 493.0810, found 493.0807.

#### 3.2.2. General Procedure for Cyclization of Acetylated Thioglycosides to Tetracyclic Conjugates **17a**–**d**–**20a**–**d**

Acetylated thioglycoside, **13a**–**d**–**16a**–**d** (0.25 mM), was suspended in 15 mL of dried MeOH and 0.9 mL MeONa solution (0.5 N) was added. The mixture was stirred at room temperature for 1 h until TLC analysis indicated complete consumption of initial thioglycoside. During the reaction, the formation of a new polar compound precipitate was also observed. The reaction mixture was acidified with 0.25 mL HCl solution (2 N) and the precipitate was filtered off, washed with water, cold MeOH, and gave high purity quinone-tioglycosidic conjugates **17a**–**d**–**20a**–**d**.

*(2R,3S,4S,4aR,12aS)-3,4-Dihydroxy-2-hydroxymethyl-7,10-dimethoxy-3,4,4a,12a-tetrahydro-2H-naphtho[2,3-b]pyrano[2,3-e][2,5]oxathiine-6,11-dione* (**17a**); yield 88 mg (86.1%), red solid, R*_f_* 0.35 (C), m.p. 332–335 °C^27^. ^1^H-NMR(700 MHz, DMSO-*d*_6_), δ: 3.30 (m, 1H, H-3), 3.47 (m, 3H, H-2, H-4a, H-13a), 3.57 (m, 1H, H-4), 3.74 (m,1H, H-13b), 3.84 (s, 3H, -OCH_3_), 3.85 (s, 3H, -OCH_3_), 4.72 (br s, 1H, C^13^OH), 4.92 (d, 1H, *J* = 8.3 Hz, H-12a), 5.37 (br s, 1H, C^3^OH), 5.58 (br s, 1H, C^4^OH), 7.52 (d, 1H, *J* = 9.6 Hz, Ar-H), 7.54 (d, 1H, *J* = 9.6 Hz, Ar-H). ^13^C-NMR (176 MHz, DMSO-*d*_6_), δ: 56.7 (-OCH_3_), 56.8 (-OCH_3_), 60.8 (C-13), 70.5 (C-3), 73.7 (C-12a), 73.9 (C-4), 79.2 (C-4a), 82.1 (C-2), 118.6, 118.9, 121.7, 122.0, 122.4, 149.7, 153.3, 153.8, 174.7, 179.9. IR (KBr): 3444, 1638, 1614, 1561, 1476, 1405, 1266, 1181, 1075, 935 cm^−1^. HRMS (ESI): *m*/*z* [M + Na]^+^ calcd. for C_18_H_18_NaO_9_S 433.0564, found 433.0562.

*(2R,3R,4S,4aR,12aS)-3,4-Dihydroxy-2-hydroxymethyl-7,10-dimethoxy-3,4,4a,12a-tetrahydro-2H-naphtho[2,3-b]pyrano[2,3-e][2,5]oxathiine-6,11-dione* (**17b**); yield 95 mg (93%), orange solid, R*_f_* 0.30 (C), m.p. > 350 °C. ^1^H-NMR(500 MHz, DMSO-*d*_6_), δ: 3.55 (m, 2H, H-13), 3.72 (t, 1H, *J* = 6.0 Hz, H-2), 3.77 (m, 2H, H-4, H-4a), 3.83 (m, 1H, H-3), 3.84 (s, 3H, -OCH_3_), 3.85 (s, 3H, -OCH_3_), 4.74 (t, 1H, *J* = 5.6 Hz, C^13^OH), 4.88 (d, 1H, *J* = 7.6 Hz, H-12a), 4.89 (d, 1H, *J* = 4.6 Hz, C^3^OH), 5.31 (d, 1H, *J* = 6.3 Hz, C^4^OH), 7.52 (d, 1H, *J* = 9.7 Hz, Ar-H), 7.55 (d, 1H, *J* = 9.7 Hz, Ar-H). ^13^C-NMR (125 MHz, DMSO-*d*_6_), δ: 56.7 (-OCH_3_), 56.8 (-OCH_3_), 60.5 (C-13), 69.4 (C-3), 70.4 (C-4), 74.4 (C-12a), 77.7 (C-4a), 80.6 (C-2), 118.6, 118.9, 121.7, 122.0, 122.4, 150.3, 153.3, 153.8, 174.4, 179.8. IR (KBr): 3487, 3289, 3016, 2968, 2935, 2838, 1649, 1611, 1581, 1562, 1476, 1434, 1407, 1384, 1349, 1325, 1267, 1213, 1196, 1180, 1110, 1082, 1056, 1041, 1019, 1008, 936, 904, 873, 824, 803, 755 cm^−1^. HRMS (ESI): *m*/*z* [M + Na]^+^ calcd. for C_18_H_18_NaO_9_S 433.0564, found 433.0565.

*(3R,4S,4aR,12aS)-3,4-Dihydroxy-7,10-dimethoxy-3,4,4a,12a-tetrahydro-2H-naphtho[2,3-b]pyrano[2,3-e][2,5]oxathiine-6,11-dione* (**17c**); yield 80 mg (83.9%), orange solid, *R_f_* 0.47 (C), m.p. 329–332 °C. ^1^H-NMR(700 MHz, DMSO-*d*_6_), δ: 3.37 (m, 1H, H-2a), 3.48 (m, 1H, H-4a), 3.52 (m, 2H, H-3, H-4), 3.84 (s, 3H, -OCH_3_), 3.85 (s, 3H, -OCH_3_), 3.91 (dd, 1H, *J* = 11.1, 4.8 Hz, H-2b), 4.85 (d, 1H, *J* = 8.1 Hz, H-12a), 5.38 (d, 1H, *J* = 4.8 Hz, C^3^OH), 5.61 (d, 1H, *J* = 5.3 Hz, C^4^OH), 7.52 (d, 1H, *J* = 9.6 Hz, Ar-H), 7.54 (d, 1H, *J* = 9.6 Hz, Ar-H). ^13^C-NMR (176 MHz, DMSO-*d*_6_), δ: 56.7 (-OCH_3_), 56.8 (-OCH_3_), 70.0 (C-3), 70.3 (C-2), 74.1 (C-4), 74.6 (C-12a), 79.2 (C-4a), 118.6, 118.9, 121.7, 122.1, 122.3, 149.7, 153.4, 153.9, 174.6, 179.8. IR (KBr): 3502, 3470, 3296, 3013, 2981, 2939, 2877, 2836, 1661, 1636, 1611, 1581, 1562, 1478, 1459, 1431, 1408, 1359, 1281, 1267, 1253, 1223, 1196, 1164, 1125, 1095, 1060, 1042, 1023, 975, 935, 887, 818, 797, 755, 718 cm^−1^. HRMS (ESI): *m*/*z* [M + Na]^+^ calcd. for C_17_H_16_NaO_8_S 403.0458, found 403.0459.

*(3S,4S,4aR,12aS)-3,4-Dihydroxy-7,10-dimethoxy-3,4,4a,12a-tetrahydro-2H-naphtho[2,3-b]pyrano[2,3-e][2,5]oxathiine-6,11-dione* (**17d**); yield 88 mg (92.3%), yellow solid, *R_f_* 0.43 (C), m.p. 317–320 °C. ^1^H-NMR(700 MHz, DMSO-*d*_6)_, δ: 3.77 (m, 3H, H-2a, H-4, H-4a), 3.84 (m, 1H, H-3), 3.85 (s, 3H, -OCH_3_), 3.86 (s, 3H, -OCH_3_), 3.88 (dd, 1H, *J* = 12.2, 1.7 Hz, H-2b), 4.82 (d, 1H, *J* = 7.7 Hz, H-12a), 5.02 (d, 1H, *J* = 4.1 Hz, C^3^OH), 5.31 (d, 1H, *J* = 6.6 Hz, C^4^OH), 7.52 (d, 1H, *J* = 9.6 Hz, Ar-H), 7.55 (d, 1H, *J* = 9.6 Hz, Ar-H). ^13^C-NMR (176 MHz, DMSO-*d*_6_), δ: 56.7 (-OCH_3_), 56.8 (-OCH_3_), 69.4 (C-3), 69.8 (C-4), 71.4 (C-2), 74.8 (C-12a), 77.6 (C-4a), 118.6, 118.9, 121.7, 122.0, 122.4, 150.2, 153.4, 153.8, 174.6, 179.8. IR (KBr): 3496, 3449, 3094, 3012, 2979, 2932, 2864, 2837, 1659, 1642, 1613, 1479, 1459, 1433, 1409, 1352, 1338, 1325, 1280, 1259, 1206, 1188, 1166, 1123, 1097, 1080, 1069, 1048, 1022, 1007, 955, 934, 910, 866, 833, 815, 800, 747 cm^−1^. HRMS (ESI, *m*/*z*): [M + Na]^+^ calcd. for C_17_H_16_NaO_8_S 403.0458, found 403.0453.

*(2R,3S,4S,4aR,12aS)-8,9-Dichloro-3,4-dihydroxy-2-hydroxymethyl-7,10-dimethoxy-3,4,4a,12a-tetrahydro-2H-naphtho[2,3-b]pyrano[2,3-e][2,5]oxathiine-6,11-dione* (**18a**); yield 100 mg (83.4%), orange solid, R*_f_* 0.51 (C), m.p. 222–224 °C^27^. ^1^H-NMR(500 MHz, DMSO-*d*_6_), δ: 3.32 (m, 1H, H-3), 3.50 (m, 2H, H-2, H-13a), 3.54 (m, 1H, H-4a), 3.60 (m, 1H, H-4), 3.75 (m, 1H, H-13b), 3.82 (s, 3H, -OCH_3_), 3.83 (s, 3H, -OCH_3_), 4.74 (t, 1H, *J* = 5.3 Hz, C^13^OH), 4.96 (d, 1H, *J* = 8.1 Hz, H-12a), 5.40 (d, 1H, *J* = 5.7 Hz, C^3^OH), 5.67 (d, 1H, *J* = 5.8 Hz, C^4^OH). ^13^C-NMR (125 MHz, DMSO-*d*_6_), δ: 60.7 (C-13), 61.5 (2 × -OCH_3_), 70.4 (C-3), 73.6 (C-12a), 73.9 (C-4), 79.3 (C-4a), 82.2 (C-2), 123.1, 123.4, 123.9, 135.1, 135.3, 149.9, 152.3, 152.9, 173.4, 178.9. IR (KBr): 3432, 2941, 1653, 1603, 1625, 1458, 1381, 1333, 1275, 1209, 1131, 1025, 951 cm^−1^. HRMS (ESI): *m*/*z* [M + Na]^+^ calcd. for C_18_H_16_Cl_2_NaO_9_S 500.9784, found 500.9784.

*(2R,3R,4S,4aR,12aS)-8,9-Dichloro-3,4-dihydroxy-2-(hydroxymethyl)-7,10-dimethoxy-3,4,4a,12a-tetrahydro-2H-naphtho[2,3-b]pyrano[2,3-e][2,5]oxathiine-6,11-dione* (**18b**); yield 98 mg (81.7%), orange solid, R*_f_* 0.47 (C), m.p. 276–279 °C. 1H-NMR(700 MHz, DMSO-*d*6), δ: 3.56 (m, 2H, H-13), 3.75 (t, 1H, *J* = 6.0 Hz, H-2), 3.79 (dd, 1H, *J* = 9.6, 3.3 Hz, H-4), 3.82 (s, 3H, -OCH_3_), 3.83 (s, 3H, -OCH_3_), 3.85 (m, 2H, H-3, H-4a), 4.80 (br s, 2H, 2 × OH), 4.92 (d, 1H, *J* = 7.9 Hz, H-12a), 5.40 (br s, 1H, OH). ^13^C-NMR (176 MHz, DMSO-*d*_6_), δ: 60.5 (C-13), 61.5 (2 × -OCH_3_), 69.5 (C-3), 70.4 (C-4), 74.3 (C-12a), 77.9 (C-4a), 80.8 (C-2), 123.2, 123.4, 123.9, 135.1, 135.2, 150.5, 152.3, 152.9, 173.4, 178.9. IR (KBr): 3413, 2940, 2855, 1652, 1603, 1542, 1524, 1457, 1380, 1336, 1274, 1236, 1207, 1121, 1090, 1024, 945, 876, 839, 805, 762 cm^−1^. HRMS (ESI): *m*/*z* [M + Na]^+^ calcd. for C_18_H_16_Cl_2_NaO_9_S 500.9784, found 500.9783.

*(3R,4S,4aR,12aS)-8,9-Dichloro-3,4-dihydroxy-7,10-dimethoxy-3,4,4a,12a-tetrahydro-2H-naphtho[2,3-b]pyrano[2,3-e][2,5]oxathiine-6,11-dione* (**18c**); yield 102 mg (90.7%), orange solid, R*_f_* 0.59 (C), m.p. 277–279 °C. 1H-NMR(700 MHz, DMSO-*d*6), δ: 3.40 (m, 1H, H-2a), 3.54 (m, 3H, H-4a, H-4, H-3), 3.81 (s, 3H, -OCH_3_), 3.83 (s, 3H, -OCH_3_), 3.93 (dd, 1H, *J* = 11.2, 4.6 Hz, H-2b), 4.90 (d, 1H, *J* = 7.5 Hz, H-12a), 5.42 (d, 1H, *J* = 4.4 Hz, C^3^OH), 5.72 (d, 1H, *J* = 4.9 Hz, C^4^OH). ^13^C-NMR (176 MHz, DMSO-*d*_6_), δ: 61.5 (2 × -OCH_3_), 69.9 (C-3), 70.4 (C-2), 74.0 (C-4), 74.5 (C-12a), 79.4 (C-4a), 123.0, 123.4, 123.8, 135.1, 135.3, 149.9, 152.3, 152.9, 173.4, 178.9. IR (KBr): 3482, 3385, 3003, 2938, 2880, 2855, 1651, 1604, 1544, 1524, 1458, 1382, 1330, 1307, 1272, 1237, 1225, 1208, 1124, 1090, 1074, 1061, 1054, 1024, 975, 948, 898, 869, 838, 816, 802, 763, 741 cm^−1^. HRMS (ESI): *m*/*z* [M + Na]^+^ calcd. for C_17_H_14_Cl_2_NaO_8_S 470.9679, found 470.9673.

*(3S,4S,4aR,12aS)-8,9-Dichloro-3,4-dihydroxy-7,10-dimethoxy-3,4,4a,12a-tetrahydro-2H-naphtho[2,3-b]pyrano[2,3-e][2,5]oxathiine-6,11-dione* (**18d**); yield 97 mg (86.3%), orange solid, R*_f_* 0.56 (C), m.p. 286–289 °C. ^1^H-NMR(500 MHz, DMSO-*d*_6_), δ: 3.80 (m, 2H, H-2a, H-4), 3.82 (s, 3H, -OCH_3_), 3.83 (s, 3H, -OCH_3_), 3.85 (m, 2H, H-3, H-4a), 3.90 (dd, 1H, *J* = 12.2, 1.7 Hz, H-2b), 4.86 (d, 1H, *J* = 8.1 Hz, H-12a), 5.05 (br s, 1H, C^3^OH), 5.40 (br s, 1H, C^4^OH). ^13^C-NMR (125 MHz, DMSO-*d*_6_), δ: 61.5 (2 × -OCH_3_), 69.4 (C-3), 69.8 (C-4), 71.5 (C-2), 74.7 (C-12a), 77.8 (C-4a), 123.2, 123.4, 123.9, 135.1, 135.2, 150.4, 152.3, 152.9, 173.4, 178.9. IR (KBr): 3391, 2980, 2940, 2856, 1654, 1603, 1543, 1525, 1458, 1381, 1334, 1305, 1272, 1237, 1211, 1128, 1092, 1048, 1026, 967, 944, 913, 876, 839, 805, 757 cm^−1^. HRMS (ESI): *m*/*z* [M + Na]^+^ calcd. for C_17_H_14_Cl_2_NaO_8_S 470.9679, found 470.9675.

*(2R,3S,4S,4aR,12aS)-3,4,10-Trihydroxy-2-hydroxymethyl-3,4,4a,12a-tetrahydro-2H-naphtho[2,3-b]pyrano[2,3-e][2,5]oxathiine-6,11-dione* (**19a**); yield 86 mg (93.7%), orange solid, R*_f_* 0.56 (C), decomposition > 342 °C. ^1^H-NMR(700 MHz, DMSO-*d*_6_), δ: 3.32 (m, 1H, H-3), 3.50 (m, 2H, H-2, H-13a), 3.60 (m, 2H, H-4, H-4a), 3.75 (m, 1H, H-13b), 4.74 (t, 1H, *J* = 5.5 Hz, C^13^OH), 5.01 (d, 1H, *J* = 7.7 Hz, H-12a), 5.41 (d, 1H, *J* = 5.8 Hz, C^3^OH), 5.69 (d, 1H, *J* = 5.7 Hz, C^4^OH), 7.30 (dd, 1H, *J* = 8.4, 0.7 Hz, H-9), 7.54 (dd, 1H, *J* = 7.4, 0.7 Hz, H-7), 7.70 (dd, 1H, *J* = 8.4, 7.4 Hz, H-8), 11.52 (s, 1H, C^10^OH). ^13^C-NMR (176 MHz, DMSO-*d*_6_), δ: 60.7 (C-13), 70.4 (C-3), 73.5 (C-12a), 73.9 (C-4), 79.4 (C-4a), 82.3 (C-2), 113.8 (C-10a), 119.2 (C-7), 122.9 (C-11a), 124.2 (C-9), 130.8 (C-6a), 136.5 (C-8), 151.0 (C-5a), 159.8 (C-10), 175.1 (C-6), 186.1 (C-11). IR (KBr): 3464, 3351, 3233, 2955, 2881, 1646, 1616, 1580, 1517, 1477, 1455, 1416, 1383, 1356, 1324, 1297, 1280, 1252, 1229, 1219, 1194, 1165, 1148, 1133, 1092, 1055, 1037, 1000, 973, 889, 863, 836, 828, 816, 755, 728 cm^−1^. HRMS (ESI): *m*/*z* [M + Na]^+^ calcd. for C_16_H_14_NaO_8_S 389.0302, found 389.0300.

*(2R,3R,4S,4aR,12aS)-3,4,10-Trihydroxy-2-hydroxymethyl-3,4,4a,12a-tetrahydro-2H-naphtho[2,3-b]pyrano-[2,3-e][2,5]oxathiine-6,11-dione* (**19b**); yield 75 mg (81.7%), dark orange solid, R*_f_* 0.52 (C), decompose above > 346 °C. ^1^H-NMR(500 MHz, DMSO-*d*_6_), δ: 3.57 (m, 2H, H-13), 3.76 (t, 1H, *J* = 6.0 Hz, H-2), 3.80 (m, 1H, H-4), 3.85 (m, 1H, H-3), 3.90 (t, 1H, *J* = 9.5, 8.0 Hz, H-4a), 4.77 (t, 1H, *J* = 5.6 Hz, C^13^OH), 4.96 (d, 1H, *J* = 4.8 Hz, C^3^OH), 4.98 (d, 1H, *J* = 8.0 Hz, H-12a), 5.43 (d, 1H, *J* = 6.8 Hz, C^4^OH), 7.30 (dd, 1H, *J* = 8.4, 0.9 Hz, H-9), 7.54 (dd, 1H, J = 7.4, 0.9 Hz, H-7), 7.70 (dd, 1H, *J* = 8.4, 7.4 Hz, H-8), 11.53 (s, 1H, C^10^OH). ^13^C-NMR (125 MHz, DMSO-*d*_6_), δ: 60.5 (C-13), 69.5 (C-3), 70.4 (C-4), 74.1 (C-12a), 78.1 (C-4a), 80.8 (C-2), 113.8 (C-10a), 119.2 (C-7), 122.9 (C-11a), 124.2 (C-9), 130.8 (C-6a), 136.5 (C-8), 151.5 (C-5a), 159.8 (C-10), 175.1 (C-6), 186.1 (C-11). IR (KBr): 3456, 3338, 2984, 2938, 2888, 1644, 1616, 1580, 1476, 1456, 1431, 1401, 1384, 1327, 1296, 1279, 1250, 1208, 1165, 1136, 1105, 1085, 1056, 1024, 979, 903, 879, 860, 833, 813, 754, 734, 690 cm^−1^. HRMS (ESI, *m*/*z*): [M + Na]^+^ calcd. for C_16_H_14_NaO_8_S 389.0302, found 389.0295.

*(3R,4S,4aR,12aS)-3,4,10-Trihydroxy-3,4,4a,12a-tetrahydro-2H-naphtho[2,3-b]pyrano[2,3-e][2,5]oxathiine-6,11-dione* (**19c**); yield 76 mg (90.1%), orange solid, *R_f_* 0.67 (C), m.p. 348–350 °C with decomposition. ^1^H-NMR(700 MHz, DMSO-*d*_6_), δ: 3.41 (m, 1H, H-2), 3.55 (m, 2H, H-3, H-4), 3.60 (m, 1H, H-4a), 3.94 (dd, 1H, *J* = 11.1, 4.7 Hz, H-2′), 4.95 (d, 1H, *J* = 8.0 Hz, C-12a), 5.42 (d, 1H, *J* = 4.7 Hz, C^3^OH), 5.73 (d, 1H, *J* = 5.3 Hz, C^4^OH), 7.30 (dd, 1H, *J* = 8.4, 0.9 Hz, H-9), 7.54 (dd, 1H, *J* = 7.4, 0.9 Hz, H-7), 7.70 (dd, 1H, *J* = 8.4, 7.4 Hz, H-8), 11.51 (s, 1H, C^10^OH). ^13^C-NMR (176 MHz, DMSO-*d*_6_), δ: 70.0 (C-3), 70.4 (C-2), 74.0 (C-4), 74.3 (C-12a), 79.4 (C-4a), 113.8 (C-10a), 119.2 (C-7), 122.8 (C-11a), 124.2 (C-9), 130.8 (C-6a), 136.6 (C-8), 151.0 (C-5a), 159.8 (C-10), 175.1 (C-6), 186.1 (C-11). IR (KBr): 3381, 3300, 2940, 2899, 2867, 1652, 1625, 1621, 1581, 1516, 1475, 1462, 1454, 1417, 1378, 1323, 1306, 1295, 1239, 1222, 1200, 1168, 1146, 1136, 1074, 1056, 1044, 1035, 1002, 976, 962, 900, 868, 833, 818, 789, 756 cm^−1^. HRMS (ESI): *m*/*z* [M + Na]^+^ calcd. for C_15_H_12_NaO_7_S 359.0196, found 359.0197.

*(3S,4S,4aR,12aS)-3,4,10-Trihydroxy-3,4,4a,12a-tetrahydro-2H-naphtho[2,3-b]pyrano[2,3-e][2,5]oxathiine-6,11-dione* (**19d**); yield 78 mg (92.5%), orange solid, *R_f_* 0.62 (C), m.p. 319–321 °C. ^1^H-NMR(500 MHz, DMSO-*d*_6_), δ: 3.79 (m, 2H, H-4, H-2a), 3.86 (m, 1H, H-3), 3.90 (m, 2H, H-4a, H-2b), 4.91 (d, 1H, *J* = 8.0 Hz, H-12a), 5.10 (d, 1H, *J* = 4.0 Hz, C^3^OH), 5.44 (d, 1H, *J* = 7.0 Hz, C^4^OH), 7.30 (dd, 1H, *J* = 8.4, 1.0 Hz, H-9), 7.53 (dd, 1H, *J* = 7.5, 1.0 Hz, H-7), 7.70 (dd, 1H, *J* = 8.4, 7.5 Hz, H-8), 11.54 (s, 1H, C^10^OH). ^13^C-NMR (125 MHz, DMSO-*d*_6_), δ: 69.4 (C-3), 69.7 (C-4), 71.5 (C-2), 74.5 (C-12a), 77.9 (C-4a), 113.8 (C-10a), 119.2 (C-7), 122.9 (C-11a), 124.2 (C-9), 130.7 (C-6a), 136.5 (C-8), 151.4 (C-5a), 159.8 (C-10), 175.1 (C-6), 186.1 (C-11). IR (KBr): 3511, 3196, 2987, 2925, 2892, 1654, 1624, 1578, 1475, 1463, 1444, 1378, 1356, 1333, 1308, 1279, 1237, 1177, 1164, 1139, 1109, 1084, 1071, 1022, 1004, 982, 938, 911, 837, 861, 835, 815, 754 cm^−1^. HRMS (ESI, *m*/*z*): [M + Na]^+^ calcd. for C_15_H_12_NaO_7_S 359.0196, found 359.0192.

*(2R,3S,4S,4aR,12aS)-3,4,7-Trihydroxy-2-(hydroxymethyl)-3,4,4a,12a-tetrahydro-2H-naphtho[2,3-b]pyrano-[2,3-e][2,5]oxathiine-6,11-dione* (**20a**); yield 80 mg (87.1%), orange solid, R*_f_* 0.56 (C), m.p. 274–276 °C. ^1^H-NMR(700 MHz, DMSO-*d*_6_), δ: 3.33 (m, 1H, H-3), 3.50 (m, 2H, H-2, H-13a), 3.60 (m, 2H, H-4, H-4a), 3.75 (m, 1H, H-13b), 4.73 (t, 1H, *J* = 5.6 Hz, C^13^OH), 5.01 (d, 1H, *J* = 7.7 Hz, H-12a), 5.41 (d, 1H, *J* = 5.7 Hz, C^3^OH), 5.68 (d, 1H, *J* = 5.7 Hz, C^4^OH), 7.32 (dd, 1H, *J* = 8.5, 0.9 Hz, H-8), 7.51 (dd, 1H, *J* = 7.4, 0.9 Hz, H-10), 7.70 (dd, 1H, *J* = 8.5, 7.4 Hz, H-9), 11.72 (s, 1H, C^7^OH). ^13^C-NMR (176 MHz, DMSO-*d*_6_), δ: 60.7 (C-13), 70.4 (C-3), 73.6 (C-12a), 73.9 (C-4), 79.2 (C-4a), 82.2 (C-2), 113.4 (C-6a), 118.7 (C-10), 124.1 (C-11a), 124.5 (C-8), 131.2 (C-10a), 136.5 (C-9), 150.0 (C-5a), 160.7 (C-7), 180.4 (C-11), 180.5 (C-6). IR (KBr): 3493, 3474, 3415, 3365, 3264, 2959, 2931, 2892, 1658, 1628, 1585, 1481, 1457, 1403, 1457, 1380, 1296, 1280, 1246, 1223, 1198, 1156, 1143, 1124, 1096, 1077, 1048, 1035, 1009, 952, 895, 869, 834, 816, 796, 747 cm^−1^. HRMS (ESI): *m*/*z* [M + Na]^+^ calcd. for C_16_H_14_NaO_8_S 389.0302, found 389.0297.

*(2R,3R,4S,4aR,12aS)-3,4,7-Trihydroxy-2-hydroxymethyl-3,4,4a,12a-tetrahydro-2H-naphtho[2,3-b]pyrano[2,3-e][2,5]oxathiine-6,11-dione* (**20b**); yield 75 mg (81.7%), orange solid, R*_f_* 0.52 (C), m.p. 293–295 °C. ^1^H-NMR(500 MHz, DMSO-*d*_6_), δ: 3.56 (m, 2H, H-13), 3.76 (t, 1H, *J* = 6.0 Hz, H-2), 3.80 (dd, 1H, *J* = 9.5, 3.2 Hz, H-4), 3.85 (dd, 1H, *J* = 3.3, 1.2 Hz, H-3), 3.89 (d, 1H, *J* = 9.5, 7.9 Hz, H-4a), 4.83 (br s, 3H, C^3^OH, C^4^OH, C^13^OH), 4.97 (d, 1H, *J* = 7.9 Hz, H-12a), 7.32 (dd, 1H, *J* = 8.5, 1.0 Hz, H-8), 7.51 (dd, 1H, *J* = 7.5, 1.0 Hz, H-10), 7.70 (dd, 1H, *J* = 8.5, 7.5 Hz, H-9), 11.72 (s, 1H, C^7^OH). ^13^C-NMR (125 MHz, DMSO-*d*_6_), δ: 60.5 (C-13), 69.5 (C-3), 70.4 (C-4), 74.2 (C-12a), 77.9 (C-4a), 80.8 (C-2), 113.4 (C-6a), 118.7 (C-10), 124.1 (C-11a), 124.5 (C-8), 131.3 (C-10a), 136.5 (C-9), 150.6 (C-5a), 160.7 (C-7), 180.4 (C-11), 180.5 (C-6). IR (KBr): 3412, 2941, 1627, 1581, 1475, 1454, 1385, 1297, 1282, 1241, 1208, 1192, 1155, 1129, 1074, 1049, 986, 954, 937, 870, 832, 813, 745, 698 cm^−1^. HRMS (ESI): *m*/*z* [M + Na]^+^ calcd. for C_16_H_14_NaO_8_S 389.0302, found 389.0296.

*(3R,4S,4aR,12aS)-3,4,7-Trihydroxy-3,4,4a,12a-tetrahydro-2H-naphtho[2,3-b]pyrano[2,3-e][2,5]oxathiine-6,11-dione* (**20c**); yield 70 mg (83.0%), orange solid, R*_f_* 0.67 (C), m.p. 334–336 °C. ^1^H-NMR(500 MHz, DMSO-*d*_6_), δ: 3.41 (m, 1H, H-2a), 3.56 (m, 3H, H-3, H-4, H-4a), 3.94 (dd, 1H, *J* = 11.0, 4.7 Hz, H-2b), 4.95 (d, 1H, *J* = 7.5 Hz, H-12a), 5.42 (d, 1H, *J* = 4.8 Hz, C^3^OH), 5.73 (d, 1H, *J* = 5.1 Hz, C^4^OH), 7.32 (dd, 1H, *J* = 8.5, 1.0 Hz, H-8), 7.51 (dd, 1H, *J* = 7.5, 1.0 Hz, H-10), 7.70 (dd, 1H, *J* = 8.5, 7.5 Hz, H-9), 11.71 (s, 1H, C^7^OH). ^13^C-NMR (125 MHz, DMSO-*d*_6_), δ: 69.4 (C-3), 69.7 (C-4), 71.5 (C-2), 74.5 (C-12a), 77.9 (C-4a), 113.8 (C-10a), 119.2 (C-7), 122.9 (C-11a), 124.2 (C-9), 130.7 (C-6a), 136.5 (C-8), 151.4 (C-5a), 159.8 (C-10), 180.4 (C-11), 180.5 (C-6). IR (KBr): 3401, 2887, 1653, 1622, 1579, 1476, 1454, 1407, 1390, 1297, 1277, 1251, 1220, 1207, 1194, 1156, 1134, 1090, 1063, 1048, 1008, 976, 952, 902, 840, 831, 818, 745 cm^−1^. HRMS (ESI): *m*/*z* [M + Na]^+^ calcd. for C_15_H_12_NaO_7_S 359.0196, found 359.0194.

*(3S,4S,4aR,12aS)-3,4,7-Trihydroxy-3,4,4a,12a-tetrahydro-2H-naphtho[2,3-b]pyrano[2,3-e][2,5]oxathiine-6,11-dione* (**20d**); yield 82 mg (97.3%), dark orange solid, *R_f_* 0.62 (C), m.p. 284–286 °C. ^1^H-NMR(500 MHz, DMSO-*d*_6_), δ: 3.79 (dd, 1H, *J* = 12.7, 3.4 Hz, H-2a), 3.80 (m, 1H, H-3), 3.86 (m, 1H, H-4), 3.89 (t, 1H, *J* = 8.0 Hz, H-4a), 3.91 (dd, 1H, *J* = 12.7, 1.8 Hz, H-2b), 4.90 (d, 1H, *J* = 8.0 Hz, H-12a), 5.08 (d, 1H, *J* = 4.0 Hz, C^3^OH), 5.41 (d, 1H, *J* = 6.9 Hz, C^4^OH), 7.32 (dd, 1H, *J* = 8.4, 1.0 Hz, H-8), 7.51 (dd, 1H, *J* = 7.5, 1.0 Hz, H-10), 7.69 (dd, 1H, *J* = 8.4, 7.5 Hz, H-9), 11.72 (s, 1H, C^7^OH). ^13^C-NMR (125 MHz, DMSO-*d*_6_), δ: 69.4 (C-3), 69.7 (C-4), 71.5 (C-2), 74.6 (C-12a), 77.7 (C-4a), 113.4 (C-6a), 118.7 (C-10), 124.1 (C-11a), 124.5 (C-8), 131.2 (C-10a), 136.4 (C-9), 150.5 (C-5a), 160.6 (C-7), 180.3 (C-11), 180.4 (C-6). IR (KBr): 3541, 3268, 3167, 2998, 2936, 1648, 1627, 1621, 1618, 1581, 1474, 1453, 1405, 1384, 1330, 1299, 1284, 1245, 1215, 1191, 1173, 1153, 1115, 1088, 1068, 1049, 1037, 1007, 950, 915, 881, 869, 841, 831, 812, 790, 744, 729 cm^−1^. HRMS (ESI): *m*/*z* [M + Na]^+^ calcd. for C_15_H_12_NaO_7_S 359.0196, found 359.0191.

### 3.3. Biology

#### 3.3.1. Cell Culture

Human adenocarcinoma cell line HeLa, mouse neuroblastoma cell line Neuro-2a, and mouse epithelial cells Jb6 Cl 41-5a were obtained from ATCC (Manassas, VA, USA). Mouse ascites Ehrlich carcinoma was provided by the N.N. Blokhin NMRCO (Ministry of Health of the Russian Federation, Moscow, Russia).

HeLa and Neuro-2a cells were cultured in DMEM medium containing 10% fetal bovine serum (FBS) (Biolot, Russia) and 1% penicillin/streptomycin (Biolot, Russia). Jb6 Cl 41-5a cells were cultivated in MEM medium containing 5% fetal bovine serum (FBS) (Biolot, Russia) and 1% penicillin/streptomycin (Biolot, Russia). All cell lines were incubated at 37 °C in a humidified atmosphere with 5% (*v*/*v*) CO_2_.

The cells of Ehrlich carcinoma were injected into the peritoneal cavity of 18–20 g albino CD-1 mice (male and female). Cells for experiments were collected 7 days after inoculation. The mice were killed by perivisceral dislocation and the ascitic fluid containing tumor cells was collected with a syringe. The cells were washed three times by centrifugation at 2000 rpm (450 g) for 5 min at 4 °C in phosphate-buffered saline (PBS; pH 7.4) followed by resuspension in RPMI-1640 culture medium with 10% FBS.

Erythrocytes were isolated from the blood of mice. Blood was taken from CD-1 mice (18–20 g). The mice were anesthetized with diethyl ether, their chests were rapidly opened, and blood was collected in cold (4 °C) 10 mM phosphate-buffered saline pH 7.4 (PBS) without an anticoagulant. Erythrocytes were washed by centrifugation with centrifuge LABOFUGE 400R (Heraeus, Germany) for 5 min 3 times (2000 rpm) in PBS using at least 10 vol. of washing solution. Then, the residue of erythrocytes were resuspended in ice cold phosphate-buffered saline (pH 7.4) to a final optical density of 1.0 at 700 nm and kept on ice.

All experiments were carried out in accordance with the EU Animals (Scientific Procedures) Act, 1986 and associated guidelines, EU Directive 2010/63/EU for animal experiments.

#### 3.3.2. Cell Viability Assay

##### MTT Assay

Stocks of substances were prepared in DMSO at a concentration of 10 mM. All studied compounds were tested in concentrations from 25 μM using twofold dilution. All tested compounds were added to the wells of the plates in a volume of 20 μL diluted in PBS (final DMSO concentration < 1%). A known cytotoxic agent—triterpene glycoside cucumarioside A_2_-2 from holothurian *Cucumaria japonica*—was used as positive control. Cells (3 × 10^4^ cells/well) were incubated with different concentrations of 1,4-NQs in a CO_2_ incubator for 24 h at 37 °C. After incubation, the medium with tested substances was replaced with 100 μL of pure medium. Then, 10 μL of MTT (Sigma-Aldrich, St. Louis, MO, USA) (thiazolyl blue tetrazolium bromide) stock solution (5 mg/mL) was added to each well and the microplate was incubated for 4 h. After that, 100 μL of SDS-HCl solution (1 g SDS/10 mL dH_2_O/17 μL 6 N HCl) was added to each well followed by incubation for 4–18 h. The absorbance of the converted dye formazan was measured using a Multiskan FC microplate photometer (Thermo Scientific, USA) at a wavelength of 570 nm [32]. The results were presented as percent of control data, and concentration required for 50% inhibition of cell viability (EC_50_) was calculated. Selectivity index (SI) is defined as the ratio of EC_50_ for normal cells and EC_50_ for tumor cell lines (SI = EC_50_ (normal cells)/EC_50_ (tumor cells)).

##### Nonspecific Esterase Activity Assay

The test solution (20 μL) and 180 μL of the cell suspension were placed into each well of a 96-well microplate (3.5 × 10^4^ cells/well). The plates were incubated in a CO_2_ incubator at 37 °C for 24 h. A stock solution of the probe fluorescein diacetate (FDA) (Sigma-Aldrich, St. Louis, MO, USA) in DMSO (1 mg/mL) was prepared. After incubation, cells were washed with PBS, 10 μL of FDA solution (50 μg/mL) was added to each well and the plate was incubated at 37 °C for 15 min, and fluorescence was measured with a Fluoroskan Ascent plate reader (Thermo Labsystems, Finland) at λ_ex_ = 485 nm and λ_em_ = 518 nm. All experiments were repeated in triplicate. Cytotoxic activity was expressed as the percent of cell viability.

##### Hemolytic Assay

Erythrocytes were used at a concentration that provided an optical density of 1.0 at 700 nm for a non-hemolyzed sample. In addition, 20 μL of a solution of test substances with a fixed concentration was added to a well of a 96-well plate containing 180 μL of the erythrocyte suspension. The erythrocyte suspension was incubated with substances for 1 h at 37 °C. After that, the optical density of the obtained solutions was measured at 700 nm and EC_50_ for hemolytic activity of each compound was calculated.

#### 3.3.3. Antimicrobial Assay

##### Microorganism Cultures

Bacteria *Bacillus cereus* ATCC 10702, *Escherichia coli* K-12, *Pseudomonas aeruginosa* ATCC 27853, and *Staphylococcus aureus* ATCC 21027 and fungus *Candida albicans* KMM 453 were obtained from the Collection of Marine Microorganisms (KMM) of the G.B. Elyakov Pacific Institute of Bioorganic Chemistry, Far East Branch, Russian Academy of Sciences.

##### Agar Diffusion Method

Test substances were dissolved in DMSO at concentrations of 1.0, 0.1, and 0.01 mg/mL. All strains were grown on trypticase-soy agar TSA (BBL) at 37 °C. After overnight incubation of each test organism in Petri dishes with the same medium, wells with a diameter of 7 mm were cut and 100 μL of the test substance solution was added. After incubation of the dishes at 37 °C for 24 h (for bacteria) and 48 h (for the fungus) at 28 °C, the diameters of the growth inhibition zones of pathogenic microorganisms (mm) were measured, including the diameter of the wells. Commercial antibacterial drugs (vancomicin and gentamicin) and antifungal (clotrimazol) drugs were used as positive controls.

##### Determination of Minimum Inhibitory Concentration (MIC)

Stocks of substances were prepared in DMSO at a concentration of 10 mM. All studied compounds were tested in concentrations from 1.5 μM to 100 μM using twofold dilution. All tested compounds were add to the wells of the plates in a volume of 20 μL diluted in PBS (DMSO concentration < 1%). The bacterial culture of *S. aureus* ATCC 21027 (Collection of Marine Microorganisms PIBOC FEB RAS) was cultured in a Petri dish at 37 °C for 24 h on solid medium with the following composition; pepton—5.0 g/L, K_2_HPO_4_—0.2 g/L, glucose—1.0 g/L, MgSO_4_—0.05 g/L, yeast extract—1 g/L, agar—16.0 g/L, and distilled water—1.0 L. The pH of the medium was adjusted to 7.2–7.4 with NaOH solution.

The antimicrobial activity of the compounds was determined by the minimum inhibitory concentration (MIC) according to the method adopted by the Clinical and Laboratory Standards Institute (CLSI). Methods for dilution antimicrobial susceptibility tests for bacteria that grow aerobically followed the approved standard—tenth edition. CLSI document M07-A10, PA: Clinical and Laboratory Standards Institute; 2015, with a slight modification of the medium. The assays were performed in 96-well microplates in appropriate broth (pepton—5.0 g/L, K_2_HPO_4_—0.2 g/L, glucose—1.0 g/L, MgSO_4_—0.05 g/L, yeast extract—1 g/L, casein hydrolyzate—2.5 g/L, distilled water—1L). The *S. aureus* bacterial suspension had a concentration of 10 × 10^9^ CFU/mL in the medium, 180 μL was then added to each well of the microplates, followed by incubation at 37 °C for 24 h. The MIC value is defined as the lowest concentration of compounds resulting in the complete inhibition of bacterial growth by measuring the absorbance at 600 nm with a microplate reader (BioTek, Winooski, Vt, USA). All experiments were performed twice in triplicate. Gentamicin was used as a positive control in concentration 1 mg/mL; 1% DMSO solution in PBS as a negative. The selectivity index for S. *aureus* has been defined as the ratio of EC_50_ for normal cells and the MIC value for the bacterial culture (SI = EC_50_/MIC).

## 4. Conclusions

In summary, a series of new tetracyclic oxathiine-fused quinone-thioglycoside conjugates based on biologically active 1,4-naphthoquinones (chloro-, hydroxy-, and methoxysubstituted) have been synthesized, characterized, and evaluated for their cytotoxic and antimicrobial activities. It was shown that the most active compounds are tetracyclic conjugates of juglone (5-hydroxy-1,4-naphthoquinone), which showed high cytotoxic activity with EC_50_ in the range of 0.3 to 0.9 μM for all cancer and noncancer cell lines. Furthermore, for the first time the antimicrobial activity for this type of compounds was evaluated by the agar diffusion method. Among the tested conjugates, the activity of juglone derivatives with a d-xylose or l-arabinose moiety and hydroxyl group at C-7 position of naphthoquinone core was comparable with antibiotics vancomicin and gentamicin against Gram-positive bacteria *S. aureus* and *B. cereus*. In liquid media, the juglone-arabinosidic tetracycles showed highest activity with a MIC of 6.25 µM against *S. aureus* strain. Thus, the positive effect of heterocyclization with mercaptosugars on cytotoxic and antimicrobial activity for a group of 1,4-naphthoquinones was shown. The effect of chloro-, hydroxy-, and metoxysubstituents on teracycles activity was also studied and a significant effect of the hydroxy group on activity was shown. It can be assumed that further modification of such tetracycles may lead to new compounds with selectivity for cancer cells and great antimicrobial activity.

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
