# Peer review of "Synthesis and Evaluation of Antimicrobial and Cytotoxic Activity of Oxathiine-Fused Quinone-Thioglucoside Conjugates of Substituted 1,4-Naphthoquinones"

_molecules, 2020, doi:10.3390/molecules25163577_

Round 1

Reviewer 1 Report

This paper describes many interesting conjugate molecules combining naphthoquinones with thio-sugars through a sulfide link.  All the new compounds have been well characterised.  The wide range of biological testing has shown some encouraging activity in the case of one particular group of compounds.  I recommend publication after the following corrections are made and the manuscript is lightly edited for English.

1.  Page 1, Title, line 3:  "Oxathiine"; also page 2, line 76.

2.  Page 2, line 72:  "Ac2O/Py"

Author Response

 Point 1:

I recommend publication after the following corrections - 

1.  Page 1, Title, line 3:  "Oxathiine"; also page 2, line 76.

2.  Page 2, line 72:  "Ac2O/Py"

Response 1: 

We have changed "oxatiine" and "oxatiane" to "OXATHIINE".

Ac2O (with subscript) is written correctly in the manuscript, but due to spesial font stile it is look strange. 

 Point 2:

I recommend publication after the manuscript is lightly edited for English

Response 2: 

We lightly edited the manuscript for English:

Line 21 – "groups" is changed to "group"

Line 106 – comma was removed after "K2CO3"

Line 134  - after [31] dot was removed

Line 145 – comma was removed after “Among them”

Line 678 – "use" is changed to "used"

Reviewer 2 Report

In this manuscript, the authors have reported the antimicrobial and cytotoxic activity of oxatiine fused quinine-thioglucoside conjugates of substituted 1,4-naphthoquinones. The manuscript represents continuation of the study previously described by this research group (references 18, 22 and 30). The manuscript is generally well written and the work presented appears sound. The experimental results are adequately documented, and I recommend publication in the Molecules after the minor corrections and modifications listed below:

The selectivity index should be included in Tables 1 and 3 (EC50 normal cell line/EC50 cancer (MIC fungal) cell lines).

Very recently a detailed study of anticancer activity of several glucose-conjugated 1,4-naphthoquinones has been published (Y. E. Sabutski was one of co-authors) in Cancers. It could be interesting to provide any explanation of mechanism of action of newly synthesized compounds.

Author Response

Point 1:

The selectivity index should be included in Tables 1 and 3 (EC50 normal cell line/EC50 cancer (MIC fungal) cell lines)

Response 1: 

We have added new table (table 2, line 157) with calculated selectivity index for cytotoxic tetracycles 18a-d - 20a-d. Also new column with SI for S. aureus was added (table 4, line 197). 

Point 2:

Very recently a detailed study of anticancer activity of several glucose-conjugated 1,4-naphthoquinones has been published (Y. E. Sabutski was one of co-authors) in Cancers. It could be interesting to provide any explanation of mechanism of action of newly synthesized compounds.

Response 2: 

Since we did't conduct detailed research of mechanism of action for newly synthesized compounds before, we can only assume that these conjugates have the same mechanism as described in Cancers (Cancers. 201911, 11, 1690. doi:10.3390/cancers11111690). Currently we only have started such studying and don't have enough information about it. 

Reviewer 3 Report

The paper describes the synthesis of oxatiine fused quinone thioglucoside conjugates of substituted 1,4-Naphthoquinones and the evaluation of their antimicrobial and cytotoxic activity. The products described are new and their characterisation is adequate.
I suggest that the paper is accepted after minor corrections are made.
Corrections:
- Editing of the english language.
- The supporting information file needs to be improved. The area in the beginning of each spectrum (close to 0 ppm) needs to be visible. Each spectrum needs to have a title and some information about the spectrum. Moreover, please provided zoomed images for all regions where the peaks are not clearly visible.

Author Response

Point 1:

Editing of the english language

Response 1:

Line 3 (title) – “oxatiine” is changed to “oxathiine”

Line 21 – “groups” is changed to “group”

Line 79 – “oxatiane” is changed to “oxathiine”

Line 106 – comma was removed after K2CO3

Line 134  - after [31] dot was removed

Line 596 – preposition “on” was changed to “in”

Line 597 - comma was removed after “dislocation”

Line 603 - comma was removed after “saline”

Line 607 - comma was removed after “700 nm”

Line 671 – article “the” is added

Line 678 – "use" is changed to "used"

Point 2:

The supporting information file needs to be improved. The area in the beginning of each spectrum (close to 0 ppm) needs to be visible. Each spectrum needs to have a title and some information about the spectrum. Moreover, please provided zoomed images for all regions where the peaks are not clearly visible.

Response 2:

We have improved supporting information file. Each NMR spectrum now has area close to 0 ppm and title with some information about it. Also we have provided zoomed images for all regions where the peaks are not clearly visible.